# ReAct Meets ActRe: Autonomous Annotation of Agent Trajectories for Contrastive Self-Training

**Zonghan Yang**[1], **Peng Li**[2*], **Ming Yan**[3], **Ji Zhang**[3], **Fei Huang**[3], **Yang Liu**[1,2,4*]

[1] Department of Computer Science and Technology, Tsinghua University, Beijing, China

[2] Institute for AI Industry Research (AIR), Tsinghua University, Beijing, China

[3] Institute of Intelligent Computing, Alibaba Group

[4] Jiangsu Collaborative Innovation Center for Language Competence, Jiangsu, China

`yangzh20@mails.tsinghua.edu.cn`

## Abstract

Language agents have demonstrated autonomous decision-making abilities by reasoning with foundation models. Recently, efforts have been made to train language agents for performance improvement, with multi-step reasoning and action trajectories as the training data. However, collecting such trajectories still requires considerable human effort, by either artificial annotation or implementations of diverse prompting frameworks. In this work, we propose $A^3T$, a framework that enables the **A**utonomous **A**nnotation of **A**gent **T**rajectories in the style of ReAct. The central role is an ActRe prompting agent, which explains the reason for an arbitrary action. When randomly sampling an external action, the ReAct-style agent could query the ActRe agent with the action to obtain its textual rationales. Novel trajectories are then synthesized by prepending the posterior reasoning from ActRe to the sampled action. In this way, the ReAct-style agent executes multiple trajectories for the failed tasks, and selects the successful ones to supplement its failed trajectory for contrastive self-training. Realized by policy gradient methods with binarized rewards, the contrastive self-training with accumulated trajectories facilitates a closed loop for multiple rounds of language agent self-improvement. We conduct experiments using QLoRA fine-tuning with the open-sourced Mistral-7B-Instruct-v0.2. In AlfWorld, the agent trained with $A^3T$ obtains a 1-shot success rate of 96%, and 100% success with 4 iterative rounds. In WebShop, the 1-shot performance of the $A^3T$ agent matches human average, and 4 rounds of iterative refinement lead to the performance approaching human experts. $A^3T$ agents significantly outperform existing techniques, including prompting with GPT-4, advanced agent frameworks, and fully fine-tuned LLMs.

## 1 Introduction

The rapid development of Large Language Models (LLMs) (OpenAI, 2023; Touvron et al., 2023; Team et al., 2023; Jiang et al., 2024) has led to the prosperity of language agents. Leveraging the ability of LLMs, language agents have demonstrated impressive performances in diverse decision-making scenarios by interacting with the environments autonomously (Wang et al., 2023; Mirchandani et al., 2023; Zheng et al., 2024; Wu et al., 2024).

Recently, increasing efforts have been made to train language agents with open-sourced LLMs. The multi-step trajectories that describe the entire task-solving process of a language agent are used as training data, which consist of environmental observations, internal reasoning texts, and external actions. The collection of such trajectories is therefore essential, which are currently categorized into two paradigms in Fig. 1 (a) and (b). The first paradigm is to leverage expert demonstrations (Yao et al., 2022). However, the expense of human labor

---

*Corresponding authors: P.L. (lipeng@air.tsinghua.edu.cn) and Y.L. (liuyang2011@tsinghua.edu.cn)

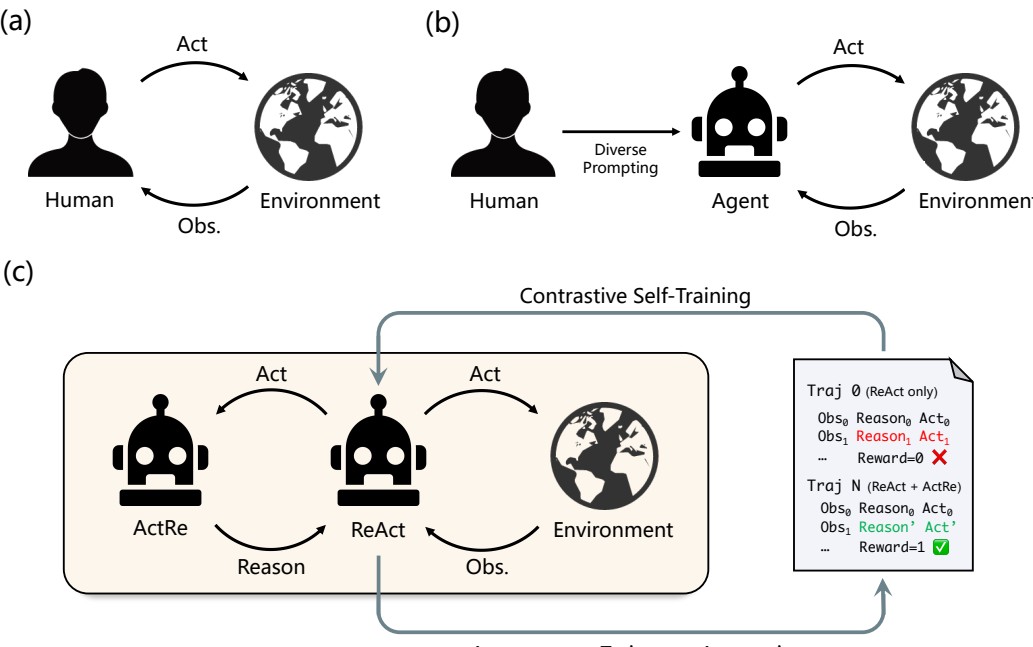

Figure 1: Upper: Two common paradigms to collect trajectories for language agents. (a) Trajectories are artificially annotated as human demonstrations. (b) Trajectories are gathered by deploying policy agents that reason and act in the language form. However, both paradigms require considerable human effort in either data annotation or different implementations of agent frameworks, thus lacking scalability in the data collection process. Lower: (c) Our $A^3T$ framework. $A^3T$ enables the **A**utonomous **A**nnotation of **A**gent **T**rajectories in ReAct style with an ActRe agent, facilitating the closed loop of contrastive self-training.

hampers the scalability of the approach. Another paradigm is implementing different agent frameworks to gather diverse trajectories with proprietary LLMs (Qin et al., 2023; Zeng et al., 2023; Chen et al., 2023; Aksitov et al., 2023). However, the exploration coverage in the training data is still upper-bounded by the full set of prompting techniques. Besides, implementing diverse agent frameworks requires considerable human efforts and proprietary LLM calls (Yang et al., 2024). To ease the data collection process in diverse scenarios, Yin et al. (2023) and Zhang et al. (2024) propose unified data formats by elucidating the comprising submodules in agent trajectories. However, as obtained by converting human-annotated data or one single defaulted prompting scheme, the agent trajectories are still limited in diversity and scalability. Considering that an environment automatically returns observations and rewards with action inputs, it should serve as an infinite data generator. While Song et al. (2024) propose an exploration-based agent framework for self-improvement, the gathered trajectories consist of only interleaved external actions and environmental observations, without textual rationales that could steer better behavior of language agents. We ask the following question: *Can a language agent autonomously gather high-quality trajectories from the external environment, with textual annotations suitable for further self-training?*

In this work, we propose $A^3T$, a framework that enables Autonomous Annotation of Agent Trajectories in the style of ReAct (Yao et al., 2023) for self-improvement with minimal human supervision. The central idea is to exploit both the in-context language ability and the decision-making ability of a language agent: To collect diverse trajectories, an agent could randomly sample external actions from the action space at arbitrary steps. However, the corresponding reason for the sampled action should be annotated for a ReAct-style agent. To facilitate this, we propose ActRe, an act-then-reason prompting agent that explains the reason for the sampled action. With ActRe, the ReAct-style agent composes extra reason-then-act trajectories for each failed task by inversely prepending the ActRe-prompted reason to the randomly sampled action. After the execution of each composed trajectory, the agent receives a terminal reward from the environment, which automatically annotates the quality of the trajectory. The gathered successful trajectories are then supplemented with the failed trajectory by the ReAct-style agent alone for contrastive self-training, where we use policy

gradient methods (Williams, 1992) with binarized rewards for LLM fine-tuning. As new agents are trained, more trajectories can be gathered and accumulated, which forms a closed loop for the self-improvement of language agents as shown in Fig. 1-(c).

We validate our $A^3T$ framework in the textual embodied environment AlfWorld (Shridhar et al., 2021) and the online shopping environment WebShop (Yao et al., 2022). We use QLoRA (Dettmers et al., 2023) to fine-tune Mistral-7B-Instruct-v0.2 (Jiang et al., 2023) in the training experiments. Experimental performances demonstrate significant improvement over state-of-the-art agent techniques: On AlfWorld, our trained agent achieves a 96% success rate in unseen scenarios with a single trial. On WebShop, the success rate of our agent reaches 49%, which matches the average human performance (50%). In the setting of iterative refinement, after four rounds of data collection and contrastive self-training, the accumulated success rate becomes 100% on AlfWorld and 54.8% on WebShop, narrowing the gap with human experts (59.6% on WebShop). $A^3T$ paves the way for agents with improved autonomy through the closed loop of self-annotation and contrastive self-training.

## 2 $A^3T$ for Closed-Loop Self-Improvement

In this section, we introduce the closed-loop self-improvement for agents facilitated by the $A^3T$ framework. The loop contains two parts: autonomous trajectory annotation with the ActRe agent (Sec. 2.1), and the contrastive self-training process (Sec. 2.2).

### 2.1 Autonomous Trajectory Annotation with ActRe

Agents are able to gather diverse trajectories by exploration. However, for language agents like ReAct, the actions are inferred by first reasoning with LLMs. When the agent randomly samples an action that differs from the self-inferred one, a modified reason is needed to compose a full reason-then-act trajectory. Yao et al. (2023) show that humans can modify the reasoning in the trajectory and prompt a ReAct-style agent for desirable actions. As humans can provide in-progress supervision, such a human-in-the-loop process still lacks scalability.

To automate the process, we propose a complementary ActRe prompting agent to synthesize the modified reasons by leveraging the in-context language ability of an LLM. ActRe inverts the causality of ReAct: while ReAct conditions the external action with a reason *a priori*, ActRe explains the reason *a posteriori* for an external action. The synergy of ActRe with ReAct facilitates the autonomous annotation of textual reasons: when the language agent randomly samples an external action, the reason for the sampled action is obtained by querying the ActRe prompting agent. The synthetic reason is then used as the condition of the sampled action for the ReAct-style agent. The progress of a trajectory is synchronized between the ReAct-style agent and the ActRe prompting agent, with the only difference in the order of intermediate reasoning and actions. The detailed workings are depicted below:

Denote $o_t$, $RS_t$, $EA_t$ as the environmental observation, internal reasoning, and external action at the $t$-th step, respectively. The trajectory of a ReAct-style agent reads:

$$..., o_t, RS_t, EA_t, o_{t+1}, RS_{t+1}, EA_{t+1}, ...$$

The synchronized ActRe prompting agent has the following trajectory:

$$..., o_t, EA_t, RS_t, o_{t+1}, EA_{t+1}, RS_{t+1}, ...$$

Now when the ReAct-style agent explores for a different external action at $t + 1$ step by changing $EA_{t+1}$ into $\widetilde{EA}_{t+1}$, the corresponding internal reasoning $RS_{t+1}$ should be altered as well. This is achieved by querying the ActRe prompting agent:

$$..., o_t, EA_t, RS_t, o_{t+1}, \widetilde{EA}_{t+1} \rightarrow \widetilde{RS}_{t+1},$$

then the synthesized $\widetilde{RS}_{t+1}$ and the sampled $\widetilde{EA}_{t+1}$ compose a new ReAct trajectory:

$$..., o_t, RS_t, EA_t, o_{t+1}, \widetilde{RS}_{t+1}, \widetilde{EA}_{t+1}.$$

It is important to note that the external action serves as the prior in our framework, rather than the posterior in ReAct. This implies that the sampling space for our framework is equivalent to the external environment and not skewed by the prior of LLM reasoning. At the end of each trajectory, the environment provides a terminal reward $R$ to indicate whether the execution is successful. The reward $R$ automatically annotates the quality of the entire trajectory. In this way, the language agent autonomously composes diverse ReAct-style trajectories without human annotation effort, paving the way for self-training.

## 2.2 Contrastive Self-Training

Language agents are trained by fine-tuning an LLM with the accumulated trajectories. While supervised fine-tuning (SFT) with high-quality data is widely adopted (Zhou et al., 2023b; Singh et al., 2023), in this work, we improve the awareness of the agent about the contrast between failed and successful trajectories in the same task with policy gradient methods (Williams, 1992). In the context of ReAct-style language agents, a gathered trajectory $\tau$ with $T$ steps reads $\tau = \{o_1, a_1, o_2, a_2, \cdots, o_t, a_T\}$, with $o_t$ in token strings representing the $t$-step environmental observation, and $a_t$ representing the textual action of the agent being either the internal reasoning $RS_t$ or the external action $EA_t$. Given a total of $M$ trajectories $\{\tau^m\}$, we maximize the following objective as the estimation of policy gradient:

$$
\begin{aligned}
\nabla_\theta J(\theta) &= \frac{1}{M} \sum_{m=1}^{M} R(\tau^m) \nabla_\theta \log p_\theta(\tau^m) \\
&= \frac{1}{M} \sum_{m=1}^{M} R(\tau^m) \sum_{t=1}^{T} \nabla_\theta \log p_\theta(o_t^m | a_{<t}^m, o_{<t}^m) + \nabla_\theta \log p_\theta(a_t^m | o_{\leq t}^m, a_{<t}^m),
\end{aligned}
\tag{1}
$$

with $R(\tau^m) \leq 1$ as the score of the trajectory $\tau^m$, and $p_\theta$ as the LLM with parameters $\theta$ to be fine-tuned. While traditional policy gradient methods omit the $p_\theta(o_t^m | a_{<t}^m, o_{<t}^m)$ world modeling part, in our work, we keep the term and tune $p_\theta$ to learn a joint model of action and world modeling. This instructs the LLM to better align with the tasks and the environment.

For the gathered trajectories in each task, we filter the composed ones that result in unsuccessful task completion. This ensures that all the failed trajectories generated solely by the agent are paired with successful trajectories in the same task, and all the successful trajectories are retained in the set. Assume that in the same task, we have $K \geq 1$ successful trajectories $\tau_s^1, \tau_s^2, \cdots \tau_s^K$ and a failed trajectory $\tau_f$. Then Eq.(1) for the $K+1$ trajectories can be structured as

$$
\begin{aligned}
&\sum_{k=1}^{K} \nabla \log p_\theta(\tau_s^k) + R(\tau_f) \nabla \log p_\theta(\tau_f) \\
&= \left(1 - \frac{1 - R(\tau_f)}{2K}\right) \sum_{k=1}^{K} \nabla \log p_\theta(\tau_s^k) + \frac{1 + R(\tau_f)}{2} \nabla \log p_\theta(\tau_f) \\
&\quad + \frac{1 - R(\tau_f)}{2K} \sum_{k=1}^{K} (\nabla \log p_\theta(\tau_s^k) - \nabla \log p_\theta(\tau_f)),
\end{aligned}
\tag{2}
$$

where we use the fact that $R(\tau_s^k) = 1$ for all $k$ as they are successful trajectories. According to Eq. (2), we have the following remarks about shaping the reward of the failed trajectory:

**Remark 1.** *When $R(\tau_f) = 0$, Eq. (2) is reduced to the objective of supervised fine-tuning with only the successful trajectories, which is equivalent to Zhou et al. (2023b) and Singh et al. (2023).*

**Remark 2.** *When $R(\tau_f) = -1$, the coefficient of the second part (supervised fine-tuning on the failed trajectory) $\nabla \log p_\theta(\tau_f)$ is zeroed. The objective becomes a weighted average of supervised fine-tuning on successful trajectories (the first part), and likelihood contrast between each pair of successful/failed trajectories (the third part).*

**Remark 3.** *When $R(\tau_f) = -1$ and $K = 1$, the coefficient of the first part (supervised fine-tuning on the successful trajectories) is zeroed out as well, leaving the objective into a single likelihood contrast (the third part) between trajectory pairs. According to Rafailov et al. (2023), this leads to poor performance because of training instability.*

| Method | Pick | Clean | Heat | Cool | Examine | PickTwo | Total |
|---|---|---|---|---|---|---|---|
| BUTLER$_8$ (Shridhar et al., 2021) | 46 | 39 | 74 | **100** | 22 | 24 | 37 |
| AgentLM* (Zeng et al., 2023) | - | - | - | - | - | - | 86 |
| LM-BUTLER (Micheli & Fleuret, 2021) | 96 | 97 | 96 | 90 | 100 | **94** | **96** |
| ReAct$_6$ (Yao et al., 2023) | 92 | 58 | 96 | 86 | 78 | 41 | 71 |
| ReAct$_6$ (Our rerun) | 66 | 87 | 86 | **100** | 88 | 64 | 83 |
| A$^3$T (Round=0) | 96 | 77 | 100 | 95 | 94 | 47 | 86 |
| A$^3$T (Round=1) | 96 | 94 | 96 | 95 | 89 | **94** | 94 |
| A$^3$T (Round=2) | **100** | **100** | **100** | 95 | 89 | 82 | **96** |
| A$^3$T (Round=3) | **100** | **100** | 91 | **100** | **100** | 71 | 95 |

Table 1: Success rate (%) on each task type of AlfWorld, with a single trial on each of the 134 unseen evaluation scenarios. "BUTLER$_8$" and "ReAct$_6$" denotes the best performance of 8/6 trials following prior work. The A$^3$T agents are trained with 600 out of the total 3,553 training tasks in AlfWorld. *: The best result reported in Zeng et al. (2023).

| Method | Iters. | Pick | Clean | Heat | Cool | Examine | PickTwo | Total |
|---|---|---|---|---|---|---|---|---|
| Reflexion (Shinn et al., 2023) | 11 | 96 | 94 | **100** | 95 | **100** | **100** | 97 |
| RAFA (Liu et al., 2023) | 8 | **100** | 97 | **100** | **100** | **100** | **100** | 99 |
| A$^3$T (Round=0) | 1 | 96 | 77 | **100** | 95 | 94 | 47 | 86 |
| A$^3$T (Round=1, accum.) | 2 | 96 | 94 | **100** | 95 | **100** | **100** | 97 |
| A$^3$T (Round=2, accum.) | 3 | **100** | **100** | **100** | 95 | **100** | **100** | 99 |
| A$^3$T (Round=3, accum.) | 4 | **100** | **100** | **100** | **100** | **100** | **100** | **100** |

Table 2: Success rate (%) on each task type of AlfWorld, with iterative refinement on each of the 134 unseen evaluation scenarios. "Iters" denotes the minimum iterations of test-time refinement required to achieve the reported results[1], and "accum." means the best reward of the trajectories accumulated since the 0-th Round for each task. After the self-training process in Round 3, the success rate of the 4-shot A$^3$T agent already reaches 100%.

In implementation, we binarize the reward of the failed trajectories with $R(\tau_f) = -1$. To address Remark 3, we let the agent collect multiple successful trajectories via diverse exploration to satisfy $K > 1$. After training, the new agent would follow Sec. 2.1 to gather more annotated trajectories. The trajectory set then continually grows as looping more rounds of data collection and agent training. For the training in each round, we use the accumulated trajectory set to fine-tune an LLM with Eq. (1). Another implementation detail is that in the initial round, we use 1-shot ReAct prompting to gather the training trajectories instead of exploration and annotation for bootstrapping. The failed trajectory for each task is directly excluded as it is not paired with sampled successful trajectories. Eq. (1) is therefore reduced to ReAct supervised fine-tuning in Yao et al. (2023) for the training in Round 0. The latter rounds leverage explored trajectories via autonomous annotation, and self-training by Eq. (1) with binarized rewards. Other details are covered in Appendix A.

## 3 Experiments

We conduct experiments on two benchmarks to valid the effectiveness of A$^3$T: the textual embodied environment AlfWorld (Shridhar et al., 2021), and the online shopping environment WebShop (Yao et al., 2022). The two benchmarks require a language agent to perform multi-step decision-making to accomplish a certain goal introduced in each task.

In A$^3$T, we loop for 4 rounds of trajectory collection and agent training, with the initial round using ReAct prompting as the bootstrap of training data. No trajectories are gathered from testing tasks for training. We use gpt-3.5-turbo-instruct-0914 to implement the initial ReAct prompting, as well as the ActRe prompting agent that helps the trajectory composition in the latter 3 rounds. We use the open-sourced Mistral-7B-Instruct-v0.2 (Jiang et al., 2023) with QLoRA (Dettmers et al., 2023) finetuning for the training experiments.

| Round | Pick | Clean | Heat | Cool | Examine | PickTwo | Total |
|-------|------|-------|------|------|---------|---------|-------|
| 0 | 76 | 85 | 92 | 85 | 86 | 77 | 82 |
| 1 | 100 | 100 | 100 | 100 | 100 | 99 | 99 |
| 2 | 100 | 100 | 100 | 100 | 100 | 100 | 100 |
| 3 | 100 | 100 | 100 | 100 | 100 | 100 | 100 |

Table 3: The success rate (%) of the accumulated trajectory set in terms of each task type in the training scenarios of AlfWorld. With the agent autonomously annotating diverse trajectories, the data quality for contrastive self-training is progressively improving.

We compare our $A^3T$ framework with multiple strong baselines, including methods like advanced prompting frameworks using GPT-4, specialized LLMs by full fine-tuning, and gpt-3.5-turbo-1106 fine-tuning. The results are reported in the following sections.

## 3.1 AlfWorld

Alfworld is a textual embodied environment where an agent needs to accomplish a high-level goal by reasoning about its situation and performing sequential low-level actuation. Covering 6 task types, the benchmark provides 3,553 tasks for training and 134 held-out tasks for unseen scenarios evaluation. We use 660 out of the 3,553 tasks for our experiments: 600 for training and 60 for validation. In each round, 40 trajectories are composed for each training task failed by the policy agent. See Appendix A for other implementation details.

Baseline methods are divided into two categories: the methods that make only a single trial in each test task, and the methods that perform iterative refinement in a test task. In the former category, we select BUTLER (Shridhar et al., 2021) with supervised training over $10^5$ expert trajectories on each task type. We also select LM-BUTLER (Micheli & Fleuret, 2021) that fine-tunes a full GPT2-medium by collecting expert demonstrations with the interactions from all the 3,553 tasks (with interleaved observations and external actions in each trajectory). We also compare with the best version of the fully fine-tuned AgentLM (Zeng et al., 2023) in the AlfWorld task (AgentLM-70B), which leverages trajectories from all 3,553 training tasks in AlfWorld and other tasks in different benchmarks. The ReAct prompting (Yao et al., 2023) is also categorized into this category, and we also rerun the method with gpt-3.5-turbo-instruct-0914, following their setting to use 6 distinct prompts and report the best performance. In the latter category, we select Reflexion (Shinn et al., 2023) that prompts GPT-3.5 to self-reflect with failed trials. We also compare with RAFA (Liu et al., 2023), a principled iterative planning framework using GPT-4 as the critic.

Tables 1 and 2 show the performance comparison of our framework. For the single trial setting, the overall success rate of our agent reaches 96% at 2-nd round and matches the prior SoTA (LM-BUTLER). However, our agent is trained with a QLoRA of 26M parameters and 600 training tasks, while LM-BUTLER is fine-tuned from a full GPT2-medium of 355M parameters and all 3,553 training tasks. Besides, our agent demonstrates constant performance improvements with the 4 rounds in the held-out seen evaluation scenarios and outperforms LM-BUTLER (Table 15 in Appendix C.1). For the iterative refinement setting, our agent obtains 100% success by accumulating the decision-making trials of all the 4 trained agents from each round. The accumulated trajectory set accounts for the significance of the performance. Table 3 shows that the success rate of the trajectories composed by the agent on the training tasks improves continually. More details are covered in Appendix C.1.

## 3.2 WebShop

WebShop is an online shopping environment where an agent needs to purchase the most suitable item according to a provided instruction. The agent should navigate through a sequence of query searches and button clicks on the website to accomplish the task.

---

[1]The refinement iterations of Reflexion and RAFA are measured in their publicized logs: `https://github.com/noahshinn/reflexion/blob/main/alfworld_runs/root/reflexion_run_logs/env_results_trial_10.json` and `https://github.com/agentification/RAFA_code/blob/main/ALFWorld/run_logs/env_results_trial_7.json`.

| Test Trial | Method | Valid. | Test. |
|---|---|---|---|
| Single | ReAct (Yao et al., 2023) | - | 66.6/40.0 |
| | ReAct (Our rerun) | 63.7/34.7 | 68.5/40.0 |
| | WebGUM (Furuta et al., 2024) | - | 67.5/45.0 |
| | AgentLM* (Zeng et al., 2023) | - | 70.8/ - |
| | $A^3T$ (Round=0) | **70.1**/41.0 | 72.4/45.4 |
| | $A^3T$ (Round=1) | 69.7/43.0 | 73.1/**49.0** |
| | $A^3T$ (Round=2) | 69.0/**43.8** | 73.0/48.0 |
| | $A^3T$ (Round=3) | 69.1/42.8 | **73.9**/**49.0** |
| Iterative | $LATS_{30}$ (Zhou et al., 2023a) | - | 75.9/38.0 |
| | $A^3T_2$ (Round=1, accum.) | 74.0/47.3 | 76.6/51.6 |
| | $A^3T_3$ (Round=2, accum.) | 75.1/49.5 | 77.8/53.4 |
| | $A^3T_4$ (Round=3, accum.) | **75.9/51.3** | **78.2/54.8** |
| - | Human (average) (Yao et al., 2022) | - | 75.5/50.0 |
| | Human (expert) (Yao et al., 2022) | - | 82.1/59.6 |

Table 4: Reward ($\times 100$) and success rate ($\times 100\%$) on the validation and the test sets in WebShop. *: The best result reported in Zeng et al. (2023). "$LATS_{30}$" or "$A^3T_4$" denotes using 30 or 4 test trials. The $A^3T$ agents are trained with 2,300 out of the total 11,587 training and validation tasks in WebShop. The single-shot $A^3T$ matches averaged human performance, while the multi-shot $A^3T$ closes the performance gap with human experts.

| | LLM | easy | hard | all |
|---|---|---|---|---|
| Prompting | GPT-3.5-Turbo-16k-0613* | 52.8 | 50.6 | 52.2 |
| | GPT-4-Turbo-0613* | 67.6 | 67.4 | 67.6 |
| | GPT-4-32k-0613* | 67.5 | 69.6 | 68.1 |
| Training | xLAM-v0.1* (Zhang et al., 2024) | 53.2 | 51.2 | 52.4 |
| | $A^3T$ (Round=0) | **74.1** | 66.5 | 72.0 |
| | $A^3T$ (Round=1) | 73.6 | 73.2 | **73.5** |
| | $A^3T$ (Round=2) | 71.6 | **74.2** | 72.3 |
| | $A^3T$ (Round=3) | 72.5 | 73.9 | 72.9 |

Table 5: The average reward ($\times 100$) in the easy/hard split for WebShop by AgentBoard (Ma et al., 2024). *: Results reported in Liu et al. (2024). The single-shot $A^3T$ model significantly surpasses prompting methods with the most capable LLMs like GPT-4.

| Round | Reward R ($\times 100$) | Success Rate (%) | %{R $\geq$ 0.75} | %{R $\geq$ 0.5} |
|---|---|---|---|---|
| 0 | 68.5 | 40.0 | 51.6 | 77.5 |
| 1 | 85.2 | 61.1 | 75.8 | 94.3 |
| 2 | 88.9 | 69.4 | 82.6 | 96.3 |
| 3 | 90.6 | 73.9 | 85.0 | 96.8 |

Table 6: The reward, success rate, and percentages of reward above thresholds 0.75 and 0.5 of the accumulated trajectory set in the training tasks of WebShop. With the agent autonomously composing diverse trajectories, the data quality is continually increasing.

WebShop provides a real-valued reward $R \in [0, 1]$, with $R = 1$ as success. The benchmark provides 11,587 tasks for training and validation, and 500 held-out tasks as testing scenarios. We use 2,700 out of the 11,587 tasks for our experiments, with 2,300 for training and 400 for validation. 20 trajectories are composed for each training task failed by our trained agent in each round. Other training details are listed in Appendix A.

Baseline methods are still divided by whether or not to perform test-time iterative refinement. For the setting of a single test trial, we compare with ReAct prompting and WebGUM (Furuta et al., 2024) by jointly fine-tuning a ViT visual encoder and a Flan-T5-XL. Recently, AgentBoard (Ma et al., 2024) offers an easy/hard split of the first 251 test tasks in WebShop for better evaluation, and Liu et al. (2024) report the benchmarked results of

| Round | Reward R ($\times$100) | Success Rate (%) | %{R $\geq$ 0.75} | %{R $\geq$ 0.5} | %OP | %SC |
|---|---|---|---|---|---|---|
| 1 | 81.6 (**-4.2%**) | 55.3 (**-9.5%**) | 69.2 | 91.3 | 92.1 | 89.9 |
| 2 | 83.0 (**-6.6%**) | 58.5 (**-15.7%**) | 71.3 | 92.2 | 96.4 | 95.0 |
| 3 | 83.3 (**-8.1%**) | 59.1 (**-20.0%**) | 71.8 | 92.4 | 98.0 | 97.2 |

Table 7: The reward, success rate, and percentages of reward above thresholds 0.75 and 0.5 of the accumulated trajectory set by the ReAct-only sampling process, as well as the calculated **OP** and **SC** metrics. The relative gaps of the rewards and success rates are computed with those achieved by ActRe in Table 6.

xLAM-v0.1 (Zhang et al., 2024) with multi-task full finetuning of Mixtral-8x7B-Instruct-v0.1 (Jiang et al., 2024). We also include xLAM-v0.1 (Zhang et al., 2024) as a single-shot baseline and report the performance comparison on AgentBoard. While LUMOS (Yin et al., 2023) shares a similar spirit with xLAM-v0.1, the WebShop task is treated as an unseen scenario in their setting. To conduct a fair comparison, we do not compare ours with LU-MOS. For the setting that allows test-time iterative refinement, Reflexion has been claimed to be ineffective in Shinn et al. (2023). We compare ours with LATS (Zhou et al., 2023a), a prompting-based language agent with multiple rounds of self-reflection and tree search.

Tables 4 and 5 demonstrate the significance of A$^3$T agents. With a single test trial, the A$^3$T agent matches averaged human performance (reward: 73.9 v.s. 75.5; success rate: 49.0% v.s. 50.0%). With 4 shots of test trials, A$^3$T achieves a 54.8% success rate, closing the gap with human expert performance (59.6%). The 1-shot A$^3$T agent also outperforms prompting with GPT-4-32k-0613 in both the easy and the hard split of WebShop from AgentBoard. Table 6 further shows the quality improvement of the accumulated trajectories across multiple rounds of A$^3$T. Case studies for annotated trajectories, as well as the dataset statistics for each round of training are reported in Appendices B and C.2, respectively.

## 4  Comparisons with ReAct-only Sampling

The A$^3$T framework relies on ActRe with action sampling for trajectory annotation. In this section, we compare our design on WebShop with the sampling process conducted by the ReAct policy agent only. As Round 0 in our framework does not involve sampling, we start the comparison with our framework since Round 1, and use gpt-3.5-turbo-instruct-0914 with temperature as 0.7 under the fair API call budget. For Rounds 2 and 3, we use the self-trained ReAct policy agents to conduct trajectory sampling and compare them with our framework. To compare the collected trajectories by the two methods, we further define two metrics for quantitative analysis:

- Outperformance Rate (**OP**) computes the percentage of training scenarios where the best trajectory gathered by ActRe has a higher reward than that by ReAct-only sampling. Let the reward of the best trajectory in the $i$-th scenario gathered by ActRe be $R^{(i)}_{\text{ActRe}}$, and that by ReAct-only be $R^{(i)}_{\text{ReAct-only}}$, With the number of training scenarios in total as $N$, we define $\textbf{OP} = \sum_{i=1}^{N} \mathbf{1}\left\{R^{(i)}_{\text{ActRe}} \geq R^{(i)}_{\text{ReAct-only}}\right\}/N$.

- Success Coverage (**SC**) calculates the percentage of training scenarios where ReAct-only sampling collects successful trajectories, and ActRe achieves success as well. Let $S_{\text{ReAct-only}}$ and $S_{\text{ActRe}}$ be the set of scenarios where ReAct-only sampling and ActRe succeed, respectively, we define $\textbf{SC} = |S_{\text{ReAct-only}} \cap S_{\text{ActRe}}|/|S_{\text{ReAct-only}}|$.

According to the results in Table 7, our framework collects the trajectories with consistently higher rewards and success rates. The results of outperformance rate and success coverage also demonstrate that ActRe obtains a better coverage of high-reward trajectories over ReAct-only sampling. This shows the superiority of our framework in terms of *environmental awareness*: since external actions are directly sampled from the environment to achieve sufficient grounding, our framework bypasses the potentially limited environmental prior by LLM reasoning. Case studies of the gathered trajectories, as well as the performance comparison on WebShop between ReAct-only and ours, are attached in Appendix C.3.

| | Training Scheme | Valid. | Test. |
|---|---|---|---|
| $A^3T$ (Round=0) | Supervised with $R = 1.0$ | **70.1**/**41.0** | 72.4/**45.4** |
| Ablated (Round=0) | Supervised with $R \geq 0.75$ | 68.2/37.5 | 71.5/43.2 |
| Ablated (Round=0) | Supervised with $R \geq 0.5$ | 68.1/37.8 | **72.5**/44.6 |
| $A^3T$ (Round=1) | PG with binarized rewards $(= 1; -1)$ | 69.7/**43.0** | 73.1/**49.0** |
| Ablated (Round=1) | Supervised with $R = 1.0$ | 69.4/42.5 | 73.2/45.6 |
| Ablated (Round=1) | Supervised with label conditions | 69.5/41.3 | 72.4/45.4 |
| Ablated (Round=1) | PG with original rewards $(= 1; \mathrm{ori})$ | 69.5/41.3 | **73.4**/47.0 |
| Ablated (Round=1) | PG with original rewards $(\geq 0.75; \mathrm{ori})$ | 69.3/42.3 | 72.2/46.6 |
| Ablated (Round=1) | PG with binarized rewards $(= 1; 0.1)$ | **70.2**/42.8 | 73.0/46.6 |

Table 8: Ablated experiments of the self-training techniques in $A^3T$ on WebShop. "PG" stands for the policy gradient technique in Eq. (1). The parenthesized suffixes represent the reward configuration for the trajectories in Eq. (1). For example, "$(= 1; -1)$" means keeping the $R = 1$ trajectories, and assign the failed non-composed trajectories with $R(\tau_f) = -1$. The training trajectory set is shared among different runs in the same round, except for reward threshold constraints other than $R = 1$.

# 5    Ablation Studies

In this section, we conduct ablation studies for the self-training techniques proposed in Section 2.2, and fine-tune the proprietary gpt-3.5-turbo-1106 for further comparisons.

## 5.1    Variants of the Self-Training Techniques

We conduct ablated experiments on WebShop, as it provides a real-valued reward ranging from 0 to 1. We first study the effect of different reward thresholds for training trajectory filtering. In $A^3T$, only the trajectories with reward $R = 1$ are used in training. We change the constraint into $R \geq 0.75$ and $R \geq 0.5$ and compare the performance of the initial round with supervised ReAct fine-tuning. According to Table 8, the best performance is still obtained with $R = 1$, which agrees with the findings of Zhou et al. (2023b).

We proceed to ablate the policy gradient technique with binarized rewards and conduct experiments for training in Round 1. The first type to be compared with is supervised fine-tuning. We implement supervised training with the successful trajectories only (namely $R(\tau_f) = 0$ in Eq. (2)). This echoes the practice adopted by Singh et al. (2023). We also prepend the trajectories with the label conditions "Success" / "Fail" in the training data and conduct Round-1 supervised training. For the comparisons with policy gradient Eq. (2), we alternatively set $R(\tau_f)$ to be the original reward provided by WebShop, or to be 0.1 following the practice of Wang et al. (2024). When using the original reward, we also include another setting by relaxing the reward threshold constraint to be $R \geq 0.75$. The comparisons are shown in Table 8. Conclusions can be drawn that: (1) Policy gradient methods lead to higher promotion in task performance than supervised training methods. This resembles the effectiveness of RLHF (Ouyang et al., 2022) on top of supervised fine-tuning. (2) The binarized rewards ($\pm 1$) used in $A^3T$ lead to a significant improvement of success rate. We leave the incorporation of advanced RL (Zhou et al., 2024) and RLAIF (Lee et al., 2023; Chen et al., 2024; Hosseini et al., 2024; Yuan et al., 2024) algorithms into $A^3T$ as future work.

## 5.2    Experiments with gpt-3.5-turbo-1106 fintuning

While all of the experiments we previously reported are conducted with Mistral-7B-Instruct-v0.2 and QLoRA finetuning, in this section, we also validate $A^3T$ with gpt-3.5-turbo-1106 finetuning, the proprietary service provided by OpenAI. As the initial trajectory set for Round-0 training in $A^3T$ is obtained by ReAct prompting with gpt-3.5-turbo-instruct-0914, the starting point for the two base LLMs is the same.

| Base LLM | Pick | Clean | Heat | Cool | Examine | PickTwo | Total |
|---|---|---|---|---|---|---|---|
| Mistral-7B-Instruct-v0.2 | 96 | 77 | **100** | **95** | **94** | 47 | **86** |
| gpt-3.5-turbo-1106 | **100** | **81** | 96 | 86 | 78 | **53** | 84 |

Table 9: Success rate (%) on each task type of AlfWorld after the Round-0 (supervised) training with both of the base LLMs.

| Base LLM | Setting | Valid. | Test. |
|---|---|---|---|
| Mistral-7B-Instruct-v0.2 | Round=0 | **70.1**/41.0 | 72.4/45.4 |
| | Round=1 | 69.7/43.0 | 73.1/**49.0** |
| gpt-3.5-turbo-1106 | Round=0 | 69.4/42.0 | **73.6**/48.0 |
| | Round=1 (supervised) | 69.7/**44.0** | 72.7/46.8 |

Table 10: Reward ($\times 100$) / success rate ($\times 100\%$) on validation and test sets of WebShop using different base LLMs. For Round-1 training with gpt-3.5-turbo-1106, we conduct supervised training with $R = 1$ filtering, as only SFT is offered in the service.

| Base LLM | Reward R ($\times 100$) | Success Rate (%) | %{R $\geq$ 0.75} | %{R $\geq$ 0.5} |
|---|---|---|---|---|
| Mistral-7B-Instruct-v0.2 | 85.2 | 61.1 | 75.8 | 94.3 |
| gpt-3.5-turbo-1106 | 85.4 | 62.5 | 76.1 | 94.3 |

Table 11: Quality comparison of the accumulated trajectory set in Round 1 for WebShop, which is composed by the policy agent with the open-sourced or the proprietary LLM.

Tables 9 and 10 report the performance comparison of Round-0 supervised training between the open-sourced and the proprietary LLMs. In AlfWorld, the performance of the QLoRA fine-tuned Mistral-7B-Instruct-v0.2 even surpasses that of the proprietary gpt-3.5-turbo-1106 fine-tuning service. In WebShop, the proprietary gpt-3.5-turbo-1106 finetuning performs better in Round-0 supervised training. We then let the two models separately compose diverse trajectories for their self-training. Because of the expense of inferring with the finetuned gpt-3.5-turbo-1106 model[2], we compose 10 trajectories for each failed training task (20 with the open-sourced LLM). Shown in Table 11, the quality of the accumulated trajectories composed by the proprietary LLM is on par with those composed by the open-sourced LLM. After Round-1 self-training, the open-sourced model achieves an even higher test success rate. This is attributed to the proprietary service providing only the supervised fine-tuning option, while also indicating the importance of contrastive fine-tuning in $A^3T$.

## 6 Conclusion

In this work, we propose $A^3T$, a framework that enables the autonomous annotation of agent trajectories in the style of ReAct for contrastive self-training. The key factor in the trajectory annotation process is the ActRe prompting agent, which produces the textual rationales given arbitrary external actions. Together with ActRe and environmental feedback, the ReAct-style agent autonomously synthesizes trajectories for self-training. In the contrastive self-training process, we leverage the policy gradient methods with binarized rewards to boost the task success rate. Extensive experiments on AlfWorld and WebShop have demonstrated the superiority of $A^3T$ over multiple strong baselines.

## Acknowledgement

This work is supported by the National Natural Science Foundation of China (No. 61925601, 62276152). We thank Kaiming Liu, An Liu, Zijun Liu, Xuanyu Lei, and many others for discussions and feedback, and Ziwei Chi for the assistance with the figures. We thank all the anonymous reviewers for their valuable suggestions.

---

[2]The pricing is listed in `https://openai.com/pricing`

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

## A  Implementation Details

We implement autonomous trajectory composition in Rounds 1-3, with Round 0 as the ReAct fine-tuning bootstrapping process. The trained agent for Rounds 1-3 also follows the style of ReAct. Shown in Yao et al. (2023), humans can modify the mistaken internal reasoning to correct the action behavior of a ReAct agent. In our work, the ActRe prompting agent replaces the human role as a rationale annotator for the ReAct policy agent. The ReAct agent decides whether to sample a novel action with probability $p$. If the action is not to be sampled, then the ReAct policy agent takes the action conditioned on internal reasoning. Otherwise, the sampled action is sent to the ActRe agent as a query for posterior explanations. In the implementation, we select $p = 0.5$. If consecutive 3 actions taken by the policy agent itself are invalid, the external action is sampled one more time with the query of ActRe for reason synthesis. In AlfWorld, we let the agent gather 40 trajectories to improve the possibility of attaining at least one successful trajectory for each failed task. In WebShop experiments, due to budgetary limits, we adopt another strategy by first forcing the policy agent to collect 3 trajectories, and then terminate when a successful trajectory is gathered, or the total number of annotated trajectories reaches 20 for a single task. The prompts for ActRe in AlfWorld and WebShop are shown in Tables 12 and 13, respectively.

As for self-training, we use QLoRA finetuning with the open-sourced Mistral-7B-Instruct-v0.2 throughout the main experiments. We set the rank to be 16 and $\alpha$ to be 32 in the QLoRA configurations. The tuned modules are q_proj and v_proj only. We use the nf4 quantization without double quantization in our experiments with QLoRA. The optimizer is set to be the 32-bit paged AdamW. This allows the experiments to be conducted on 2-4 NVIDIA 3090 GPUs with 24GB memory or 2 NVIDIA A100 GPUs with 40GB memory. We set the learning rate as 1e-4, and train 10 epochs for the initial Round 0 (ReAct supervised fine-tuning with successful trajectories). In Rounds 1-3, we fine-tune the LLM for 6 epochs with Eq. 1 ($R(\tau_f) = -1$) and conduct checkpoint averaging over the entire training process.

| The ActRe in-context prompt for AlfWorld |
| --- |
| Observation: Your task is to: put some spraybottle on toilet. You are in the middle of a room. Looking quickly around you, you see a cabinet 4, a cabinet 3, a cabinet 2, a cabinet 1, a countertop 1, a garbagecan 1, a handtowelholder 2, a handtowelholder 1, a sinkbasin 2, a sinkbasin 1, a toilet 1, a toiletpaperhanger 1, and a towelholder 1. |
| Action: think[To solve the task, I need to find and take a sparybottle, then put it on toilet.] |
| Observation: OK. |
| Action: act[go to cabinet 1] |
| Reason for the action: [First I need to find a spraybottle. A spraybottle is more likely to appear in cabinet (1-4), countertop (1), toilet (1), sinkbasin (1-2), garbagecan (1). I can check one by one, starting with cabinet 1.] |
| Observation: On the cabinet 1, you see a cloth 1, a soapbar 1, a soapbottle 1. |
| Action: act[go to cabinet 2] |
| Observation: The cabinet 2 is closed. |
| Action: act[open cabinet 2] |
| Observation: You open the cabinet 2. The cabinet 2 is open. In it, you see a candle 1, and a spraybottle 2. |
| Action: act[take spraybottle 2 from cabinet 2] |
| Reason for the action: [Now I find a spraybottle (2). Next, I need to take it.] |
| Observation: You pick up the spraybottle 2 from the cabinet 2. |
| Action: act[go to toilet 1] |
| Reason for the action: [Now I take a spraybottle (2). Next, I need to put it in/on toilet 1.] |
| Observation: On the toilet 1, you see a soapbottle 2. |
| Action: act[put spraybottle 2 in/on toilet 1] |
| Observation: You put the spraybottle 2 in/on the toilet 1. |

Table 12: The ActRe prompt for the AlfWorld tasks. ActRe inverts the causality of ReAct, and provides the posterior reason for the external action.

---

**The ActRe in-context prompt for WebShop**

---

Webshop
Instruction:
i would like a 3 ounce bottle of bright citrus deodorant for sensitive skin,
and price lower than 50.00 dollars
[Search]

Action: search[3 ounce bright citrus deodorant sensitive skin]
Observation:
[Back to Search]
Page 1 (Total results: 50)
[Next >]
[B078GWRC1J]
Bright Citrus Deodorant by Earth Mama | Natural and Safe for Sensitive Skin,
Pregnancy and Breastfeeding, Contains Organic Calendula 3-Ounce
$10.99
[B078GTKVXY]
Ginger Fresh Deodorant by Earth Mama | Natural and Safe for Sensitive Skin,
Pregnancy and Breastfeeding, Contains Organic Calendula 3-Ounce
$10.99
[B08KBVJ4XN]
Barrel and Oak - Aluminum-Free Deodorant, Deodorant for Men, Essential
Oil-Based Scent, 24-Hour Odor Protection, Cedar & Patchouli Blend, Gentle on
Sensitive Skin (Mountain Sage, 2.7 oz, 2-Pack)
$15.95

Action: click[B078GWRC1J]
Reason for the action: [B078GWRC1J and B078GTKVXY are bright citrus deodorant
less then 50 dollars. I can check B078GWRC1J first.]
Observation:
[Back to Search]
[< Prev]
scent [assorted scents][bright citrus][calming lavender][ginger fresh][simply non-scents]
size [travel set (4-pack)][3 ounce (pack of 1)][3-ounce (2-pack)]
Bright Citrus Deodorant by Earth Mama | Natural and Safe for Sensitive Skin,
Pregnancy and Breastfeeding, Contains Organic Calendula 3-Ounce
Price: $10.99
Rating: N.A.
[Description]
[Features]
[Reviews]
[Buy Now]

Action: click[bright citrus]
Reason for the action: [For 3 ounce bottle of bright citrus deodorant for sensitive skin,
the item has options 'bright citrus' and '3 ounce (pack of 1)' and seems good to buy.]
Observation: You have clicked bright citrus.

Action: click[3 ounce (pack of 1)]
Reason for the action: [For 3 ounce bottle of bright citrus deodorant for sensitive skin,
the item has options 'bright citrus' and '3 ounce (pack of 1)' and seems good to buy.]
Observation: You have clicked 3 ounce (pack of 1).
Action: click[Buy Now]
Reason for the action: [For 3 ounce bottle of bright citrus deodorant for sensitive skin,
the item has options 'bright citrus' and '3 ounce (pack of 1)' and seems good to buy.]

---

Table 13: The ActRe prompt for the WebShop tasks. ActRe inverts the causality of ReAct, and provides the posterior reason for the external action.

# B  Case Studies of the Composed Trajectories

Figs. 2, 3, 4, and 5 show several cases of trajectories composed with ActRe on WebShop.

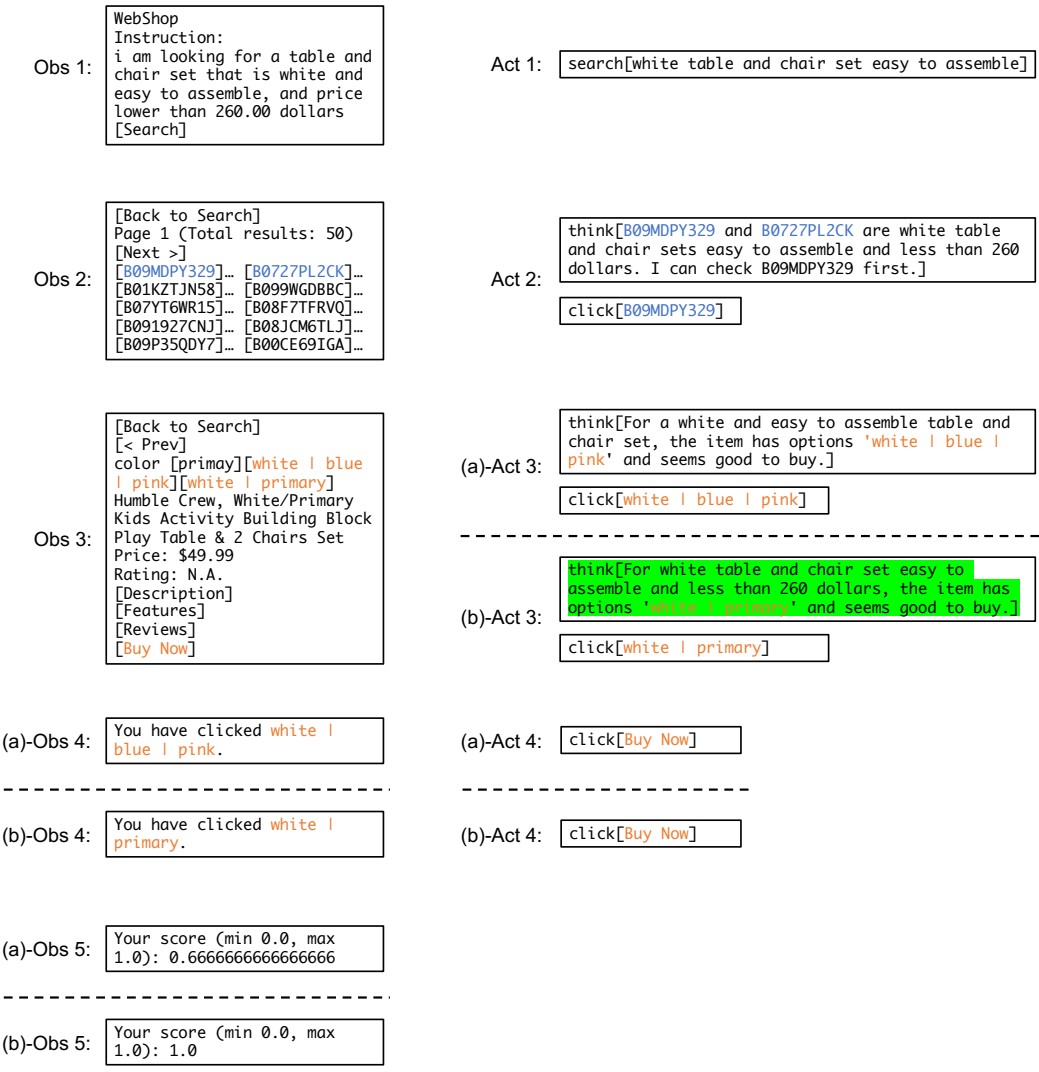

Figure 2: Trajectory comparison on the 518-th task of WebShop. (a) the failed trajectory by the trained agent at the 0-th Round; (b) the composed trajectory assisted with ActRe. The trained policy agent fails to choose the correct option in the item content page. Success is obtained by clicking "[white | primary]", and ActRe annotates for the sampled action.

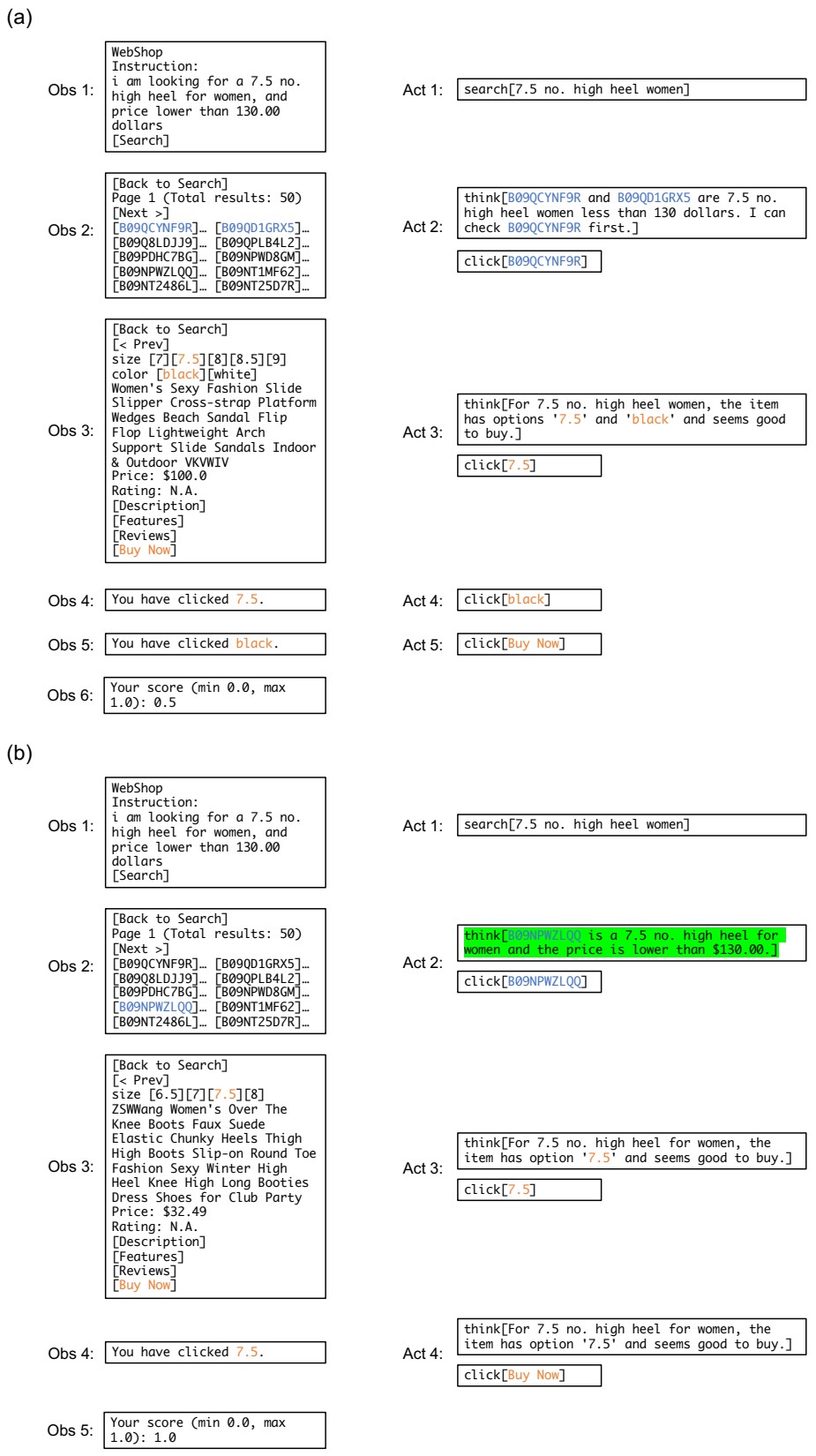

Figure 3: Trajectory comparison on the 512-th task of WebShop. (a) the failed trajectory by the trained agent at the 0-th Round; (b) the annotated trajectory assisted with ActRe. The action of clicking the 7-th item "[B09QCYNF9R]" is explored in the new trajectory, and the reason highlighted in green is generated by ActRe.

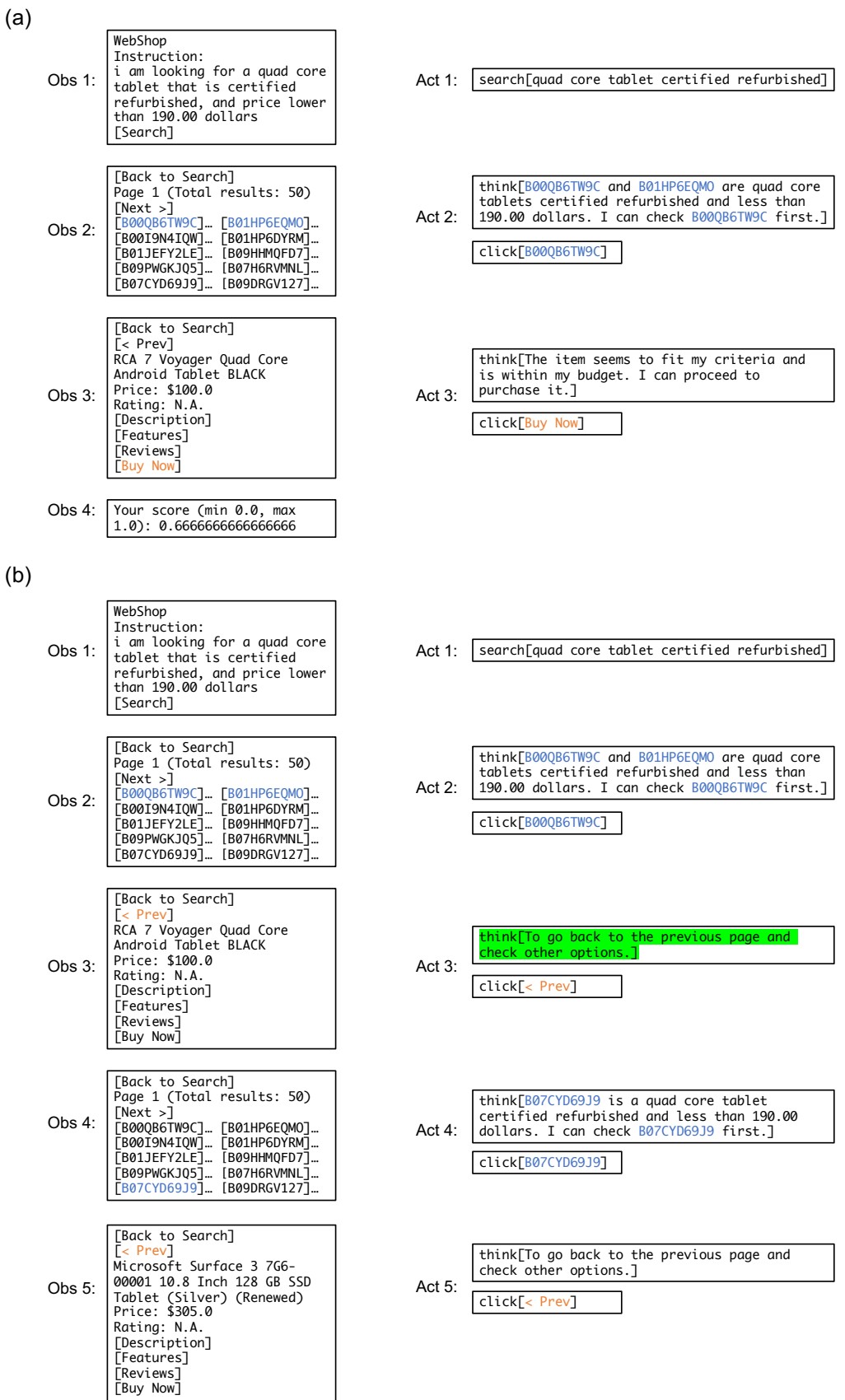

Figure 4: Trajectory comparison on the 754-th task of WebShop. (a) the failed trajectory by the trained agent at the 0-th Round; (b) the annotated trajectory assisted with ActRe. The trajectory (b) is continued in Fig. 5.

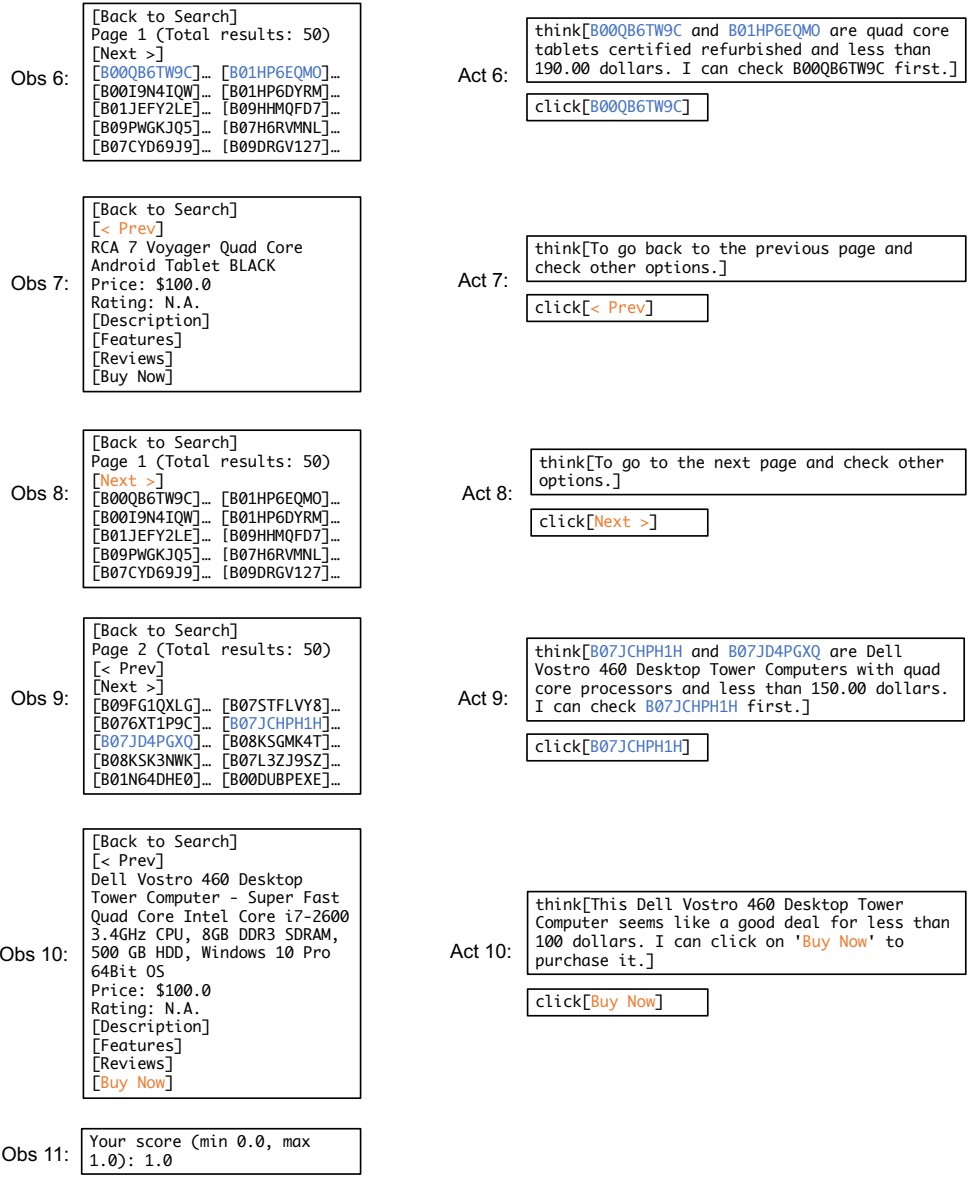

Figure 5: The second part of the composed trajectory on the 754-th task of WebShop. This case demonstrates some sophisticated behavior achieved by the synergy of the ReAct-style policy agent and the ActRe prompting agent. The policy agent alone terminates the trajectory by directing purchasing the first item ([B00QB6TW9C]) on the first page, obtaining an imperfect reward of 0.6667. By the randomly sampling actions and querying the ActRe agent to synthesize the reasons, the annotated trajectory shows that the policy agent initially clicks the first item on Page 1 as well, but then goes back to seek for other options (the reason in Act 3 highlighted in green is the first synthetic reason by ActRe). Then the agent clicks the 9-th item ([B07CYD69J9]) on Page 1, but then goes back and clicks the 1-st item, and returns to the search results of Page 1 once again. After that, the agent chooses to go to the next page, and then selects the 4-th item ([B07JCHPH1H]) and finally makes a purchase. This item results in a perfect match with the provided instruction.

## C  Additional Experimental Results

### C.1  AlfWorld

Table 14 covers the detailed task statistics in each split for our experiments.

| | Pick | Clean | Heat | Cool | Examine | PickTwo | Total |
|---|---|---|---|---|---|---|---|
| Train | 135 | 98 | 74 | 92 | 59 | 142 | 600 |
| Valid. | 12 | 10 | 7 | 14 | 8 | 9 | 60 |
| Test* (held out, seen) | 35 | 27 | 16 | 25 | 13 | 24 | 140 |
| Test (held out, unseen) | 24 | 31 | 23 | 21 | 18 | 17 | 134 |

Table 14: The statistics of tasks used in training, validation, and testing. *: For all experimental results except for Table 15, we report the success rate on the 134 held-out unseen test scenarios following the settings in previous work.

Tables 15 and 16 exhibit more experimental results on the 140 held-out seen testing tasks (official split) and the 60 validation tasks (our use).

| Method | Pick | Clean | Heat | Cool | Examine | PickTwo | Total |
|---|---|---|---|---|---|---|---|
| LM-BUTLER (Micheli & Fleuret, 2021) | 97 | 89 | 100 | 80 | 77 | **92** | 90 |
| A$^3$T (Round=0) | 86 | 67 | 94 | 80 | 85 | 75 | 80 |
| A$^3$T (Round=1) | 91 | 89 | 88 | **96** | 92 | 67 | 87 |
| A$^3$T (Round=2) | 97 | 89 | **100** | 88 | 92 | 79 | 91 |
| A$^3$T (Round=3) | **100** | **96** | **100** | **96** | 77 | 79 | **93** |

Table 15: Success rate (%) on each task type of AlfWorld, with a single trial on each of the 140 in-distribution test scenarios. Our agents outperform LM-BUTLER on this test split.

| Method | Pick | Clean | Heat | Cool | Examine | PickTwo | Total |
|---|---|---|---|---|---|---|---|
| A$^3$T (Round=0) | 11/12 | 9/10 | 5/7 | 11/14 | 7/8 | 9/9 | 52/60 |
| A$^3$T (Round=1) | 11/12 | 10/10 | 6/7 | 13/14 | 7/8 | 8/9 | 55/60 |
| A$^3$T (Round=2) | 12/12 | 10/10 | 7/7 | 13/14 | 8/8 | 8/9 | 58/60 |
| A$^3$T (Round=3) | 12/12 | 10/10 | 7/7 | 12/14 | 8/8 | 7/9 | 56/60 |

Table 16: The number of our 1-shot successful / all tasks of each type in our validation split on AlfWorld.

Table 17 shows the sentence statistics of the training datasets for each round of LLM finetuning on AlfWorld. The low percentage of the "failed" sentences (which are assigned with $R(\tau_f) = -1$) in the total training set empirically satisfies the constraint of $K > 1$ in Remark 3 of Section 2.2.

| Round | #Total | #Failed | #Failed/#Total (%) |
|---|---|---|---|
| 0 | 2,130 | 0 | 0.0 |
| 1 | 2,669 | 99 | 3.7 |
| 2 | 4,066 | 177 | 4.4 |
| 3 | 4,995 | 219 | 4.3 |

Table 17: The sentence statistics of the training datasets for LLM finetuning in each round of our trajectory collection and self-training on AlfWorld. The training set contains 0 sentences with $-1$ weights for Round 0, as Round 0 performs supervised fine-tuning with ReAct prompting trajectories as bootstrapping. The details of trajectory collection are covered in Appendix A.

## C.2  WebShop

Table 18 shows the sentence statistics of the training datasets for each round of LLM finetuning on WebShop. The percentage of the "failed" sentences is about 10%, which still satisfies the constraint of $K > 1$ in Remark 3 of Section 2.2 empirically with stable training.

| Round | #Total | #Failed | #Failed/#Total (%) |
|---|---|---|---|
| 0 | 981 | 0 | 0.0 |
| 1 | 3,431 | 336 | 9.8 |
| 2 | 5,719 | 694 | 12.1 |
| 3 | 8,550 | 1,122 | 13.1 |

Table 18: The sentence statistics of the training datasets for Mistral-7B-Instruct-v0.2 fine-tuning in each round of our trajectory collection and self-training on WebShop. Round 0 corresponds with supervised fine-tuning with ReAct trajectories, and thus uses no failed sentences. The details of trajectory collection are covered in Appendix A.

## C.3  Comparisons with ReAct-only Self-Training

Table 19 shows the comparison of performances on WebShop by ReAct-only self-training and our framework.

| Test Trial | Method | Round | Valid. | Test. |
|---|---|---|---|---|
| Single | ReAct-only | 0 | 69.1/43.2 | 72.4/45.8 |
| | | 1 | 69.2/43.2 | 73.1/47.0 |
| | | 2 | 70.0/43.5 | **74.3**/47.8 |
| | | 3 | **70.5**/**44.0** | 73.9/47.2 |
| | Ours | 0 | **70.1**/41.0 | 72.4/45.4 |
| | | 1 | 69.7/43.0 | 73.1/**49.0** |
| | | 2 | 69.0/**43.8** | 73.0/48.0 |
| | | 3 | 69.1/42.8 | **73.9**/**49.0** |
| Iterative | ReAct-only | 1 | 73.9/49.7 | 76.1/50.4 |
| | | 2 | 75.8/52.0 | 77.4/52.6 |
| | | 3 | 76.6/53.5 | 78.0/53.4 |
| | Ours | 1 | 74.0/47.3 | 76.6/51.6 |
| | | 2 | 75.1/49.5 | 77.8/53.4 |
| | | 3 | 75.9/51.3 | **78.2**/**54.8** |

Table 19: Performance comparisons between ReAct-only self-training and our framework on WebShop. Similar to Table 4, reward ($\times 100$) and success rate ($\times 100$%) on the validation and the test sets are reported.

According to the results, the performance of the ReAct-only method becomes better on the validation scenarios across different rounds, demonstrating the effectiveness of the contrastive self-training method used to train the models in Rounds 1$\tilde{3}$. However, it is shown that our framework achieves the better success rate on test scenarios. Furthermore, the consistency of outperformance is further demonstrated in the iterative refinement evaluation setting. This suggests better generalization, which might be attributed to a more sufficient coverage of high-quality trajectories gathered by our framework.

**Case Study**. The ReAct agent trained by our framework in Round 1 demonstrates the autonomous behaviour listed in Table 20.

This case demonstrates the complex reasoning ability of returning to the previous page and selecting another item. Such behavior is fostered thanks to the ActRe-gathered training data, which results in a final reward of 0.25. By comparison, all the SFT baselines (Round=0) and the agents with ReAct-only self-training achieve 0 reward at this test case. By comparing the gathered trajectories of both our framework and ReAct-only sampling, we found that

Table 20: The list of actions taken by our ReAct agent trained in Round 1 at test case #328.

search[loveseat flat packed wood finish]
→ think[B0855K1GCJ and B07J4S5JMS are flat packed wood finish loveseats less than 240 dollars.
        I can check B0855K1GCJ first.]
→ click[B0855K1GCJ] (the first item)
→ think[For a flat packed wood finish loveseat, the item has options 'flat packed' and 'wood finish'
        but does not seem to have those specifications. I can go back to the previous page and check
        other options.]
→ click[< Prev]
→ think[B07J4S5JMS is a flat packed wood finish loveseat less than 240 dollars. I can check
        B07J4S5JMS first.]
→ click[B07J4S5JMS] (the second item)
→ think[For a flat packed wood finish loveseat, the item has options '4"' and 'dark walnut finish'
        and seems good to buy.]
→ click[4"]
→ click[dark walnut finish]
→ click[Buy Now]

the "click[< Prev]" action in WebShop does not exist in the successful trajectories gathered by ReAct-only sampling in our runs, while such action has been covered in the successful trajectories annotated by our framework. The case study further supports the superiority of our framework in terms of environmental awareness, as the agent is more sufficiently grounded by directly sampling external actions for trajectory annotation, without the limitation of LLM internal reasoning priors of the external world.

