# OpenReview forum: "ReAct Meets ActRe: Autonomous Annotation of Agent Trajectories for Contrastive Self-Training"
_colmweb.org/COLM/2024/Conference — COLM_

### Official Review · Reviewer_QZXM · 2024-05-11

**Rating:** 4
**Confidence:** 3
**Ethics Flag:** 1

**Summary:**

The paper proposes a self-training method for language agents. They propose to gather new reasoning-action trajectories from an agent by first sampling an action and then generating a rationale given the sampled action at each step. Then, they train models on both successful and unsuccessful trajectories with reinforcement learning and investigate designing rewards for these trajectories. Results on AlfWorld and WebShop demonstrate the effectiveness of the method.

**Reasons To Accept:**

1. They propose a plausible way to generate self-training data for language agents.
2. They base most of their models on strong open-sourced models, which adds credibility to the results and encourages reproducibility of the work.
3. The paper is well-written.

**Reasons To Reject:**

1. Since ActRe is one of their claimed contributions, I would expect 1) better motivations and 2) more ablations and analyses on this. The current motivation of ActRe is not well-supported by the experiments, it is possible that ReAct can sample different reasoning trials that lead to diverse actions as well. In addition, ablations and analyses on this should be added. For example, self-training with ReAct-style sampling only is an important baseline that should be evaluated in all of their evaluation settings. Also, because prior work [1] demonstrates that combining different sampling strategies can lead to better performance, more discussion on previous work and more comparisons are required.
2. I do not see much difference between their contrastive self-training method with REINFORCE. I appreciate the efforts in designing the rewards and sampling strategies, but there seems to be a lack of methodological contributions.
3. While the paper mentions that for the previous work, their agent trajectories are limited in diversity and scalability, I do not see they directly solve such limitations in the experiments.


[1] Chen et al., FireAct: Toward Language Agent Fine-tuning, 2023.

---

> ### Author Rebuttal · Authors · 2024-05-31
>
> Thank you for your suggestions!
>
> We conduct self-training experiments with ReAct-only sampling on WebShop.
>
> The major finding from the ablation study is that ReAct-only sampling fails to cover all the actions in the environment, *e.g.*, the `click[< Prev]` action in WebShop is never performed by ReAct-only sampling. This leads to better motivation for ActRe: ***our framework directly samples the actions from the environment to achieve sufficient grounding, making up for the limitation of LLMs in environmental awareness when reasoning***. The following experiments support this motivation:
>
> Quantitative comparisons:
> - For the comparison of collected trajectories, under the fair settings of API call budgets, ReAct-only sampling obtains Reward=$81.6$ and Success Rate $=55.3$%, which is lower than those by ActRe (Reward **R**=$85.2$ and Success Rate **SR**$=61.1$% in Table 6).
> - For the comparison of performance after training:
>   - Using SFT with trajectories via ReAct-only sampling to train ReAct agents leads to **R**/**SR**=$72.4$/$45.8$% on test tasks, while ours in Round=0 achieves **R**/**SR**=$72.4$/$45.4$% on test tasks. This suggests that ReAct-only sampling could provide a better initial trajectory set at the cost of more API calls.
>   - Using contrastive self-training with trajectories via ReAct-only sampling to train ReAct agents leads to **R**/**SR**=$73.1$/$47.0$% on test tasks, while ours in Round=1 achieves **R**/**SR**=$73.1$/$49.0$%. This suggests the effectiveness of both ActRe sampling and contrastive self-training.
>
> Case study:
>
> - The ReAct agent trained by our framework in Round=1 autonomously conducts the following actions in the #328 testing task: click the first item --> `click[< Prev]` according to its own reasoning --> click the second item --> purchase, which shows its more complex reasoning ability thanks to the ActRe-gathered training data, and results in a final reward of 0.25. By comparison, all the SFT baselines (Round=0) and the agents trained with ReAct-only sampling achieve 0 reward.
>
> *Other Weaknesses*: While FireAct suggests also incorporating Reflexion, Reflexion does not yield improved performance on WebShop because of the environmental complexity (indicated by the Reflexion paper in Appendix B.1). We also leave more advanced RL techniques for future work.
>
> While the rebuttal length is limited, We will add more discussions to the paper, and are willing to address your concerns more thoroughly during the next period.

---

> > ### Author Response · Authors · 2024-05-31
> > **Follow-up clarification about the limitations of ReAct-only sampling**
> >
> > We'd like to add further clarification about the limitations of ReAct-only sampling, especially to rectify the potentially overclaiming statement of "`click[< Prev]` *action in WebShop is never performed by ReAct-only sampling*" in our rebuttal.
> >
> > The statement was drawn by observing the set of *successful* trajectories obtained by ReAct-only sampling: The `click[< Prev]` action does not exist in this set. When observing all the trajectories gathered by ReAct-only sampling, we indeed find that the `click[< Prev]` action has been performed. However, in our runs, all the trajectories containing the `click[< Prev]` action do not reach the full reward. Compared with the trajectory set gathered by ActRe, the ReAct-only sampling falls short of environmental understanding in terms of the possibility of `click[< Prev]` leading to success. Therefore, we'd like to update the statement in our rebuttal to "`click[< Prev]` *action in WebShop does not exist in the successful trajectories gathered by ReAct-only sampling in our runs*".
> >
> > Following the ablation studies in Section 4.1, the $R=1$ trajectories in the training set do not cover the `click[< Prev]` action for ReAct-only sampling. On the contrary, the `click[< Prev]` action often appears in the successful trajectories set with ActRe sampling. This explains the case study shown in our rebuttal.
> >
> > The motivation for ActRe proposed in our rebuttal still holds: Our framework better grounds the agent by rectifying the policy of the LLM itself with uniform action sampling from the environment when exploring the training scenarios. Such practice shares similar intuition with optimistic exploration for the less-chosen actions, as we complement additional exploration probabilities to them.

---

> > ### Comment · Reviewer_QZXM · 2024-06-03
> >
> > Thank you for your response!
> >
> > I appreciate the additional experiments, but I would expect more results from the ReAct-only baseline as mentioned in my review. I understand that this is difficult to complete during the short rebuttal period, but I do believe this is an important baseline and comparisons should be made in most of their settings.
> >
> > The added case study is a good starting point to support the claimed motivation, yet the results are relatively preliminary and more quantitative and qualitative results can be added.
> >
> > In summary, I like the provided experiments, but I would expect more of them for the paper to be published. I'm also curious about how the authors address my second and third points in the weakness section.

---

> > > ### Author Response · Authors · 2024-06-06
> > > **Thank you for your response! (1/2)**
> > >
> > > Thank you for providing us with the discussion opportunity!
> > >
> > > We are glad to know that you appreciate the experiments we added. We have been trying our best to run additional experiments to cover all the evaluation settings in the main experiments of WebShop. In the following, we provide the full comparison between ReAct-only self-training and our framework (A$^3$T with ActRe) following Tables 4 (task performance) and 6 (data quality) in our paper. For a fair comparison, we use the "SFT with initial ReAct-only sampling" model in our previous rebuttal as the Round=0 for ReAct-only, and "contrastive self-training with initial ReAct-only sampling" model in our earlier rebuttal as the Round=1 for ReAct-only. We have proceeded to use the Round=1 trained model to do ReAct-only sampling on the training scenarios, and use the accumulated trajectories to train the Round=2 ReAct-only model with contrastive self-training; The same goes for the Round-3 ReAct-only model.
> > >
> > > ### Supporting the claimed motivation
> > > ___
> > >
> > > We start with the augmented Table 6, which compares the quality of the accumulated data between our framework and ReAct-only sampling. As the initial ReAct-only sampling was conducted with fair comparison to the sampling with ActRe in Round 1, we omit Round 0 of ours in the following Table.
> > >
> > > | Round | Method     | Reward R (\*100) | Success Rate (%) | %{R$\ge$0.75} | %{R$\ge$0.5} |
> > > |-------|------------|-----------------|------------------|------------|------------------|
> > > | 1     | ReAct-only | 81.6            | 55.3             | 69.2       | 91.3             |
> > > | 1     | Ours       | 85.2            | 61.1             | 75.8       | 94.3             |
> > > |
> > > | 2     | ReAct-only | 83.0            | 58.5             | 71.3       | 92.2             |
> > > | 2     | Ours       | 88.9            | 69.4             | 82.6       | 96.3             |
> > > |
> > > | 3     | ReAct-only | 83.3            | 59.1             | 71.8       | 92.4             |
> > > | 3     | Ours       | 90.6            | 73.9             | 85.0       | 96.8             |
> > >
> > > According to the results, our framework collects the trajectories with consistently higher rewards and success rate. These quantitative results support our motivation, as the successful / high-reward trajectories could be collected only when the agent system is grounded in the environment with better awareness, and our ActRe has gathered better trajectories in this sense than the ReAct-only method that relies only on the internal reasoning of LLMs.
> > >
> > > We also define two scenario-wise comparison metrics to show the advantage of ActRe in data collection:
> > >
> > > - **Outperformance Rate** computes for the percentage of training scenarios where the best trajectory gathered by ActRe has a higher reward than that by ReAct-only sampling. Let the reward of the the best trajectory in the $i$-th scenario gathered by ActRe be $R_{\mathrm{ActRe}}^{(i)}$ , and that by ReAct-only be $R_{\mathrm{ReAct-only}}^{(i)}$. With the number of training scenarios in total as $N$, we define
> > >
> > > $$\mathrm{Outperformance~Rate} = \frac{{\sum_{i=1}^{N} \mathbf{1}} \\{R_{\mathrm{ActRe}}^{(i)} \ge R_{\mathrm{ReAct-only}}^{(i)} \\}}{N}$$
> > >
> > > - **Success Coverage** calculates the percentage of training scenarios where ReAct-only sampling collects successful trajectories, and ActRe achieves success as well. Let $S_{\rm ReAct-only}$ be the set of scenarios where ReAct-only sampling succeeds, and $S_{\rm ActRe}$ goes similar for ActRe, we define
> > >
> > > $$\mathrm{Success~Coverage} = \frac{|S_{\rm ActRe} \cap S_{\rm ReAct-only}|}{|S_{\rm ReAct-only}|}$$
> > >
> > > Computing the two metrics for the accumulated trajectories of all rounds, we have the following table:
> > >
> > > | Round | Outperformance Rate | Success Coverage |
> > > |-------|---------------------|------------------|
> > > | 1     | 92.1                | 89.9             |
> > > | 2     | 96.4                | 95.0             |
> > > | 3     | 98.0                | 97.2             |
> > >
> > > This additionally demonstrates that ActRe obtains a better coverage of high-reward trajectories over ReAct-only sampling, showing its superiority of environmental awareness and supporting our motivation.

---

> > > > ### Author Response · Authors · 2024-06-06
> > > > **Thank you for your response! (2/2)**
> > > >
> > > > ### More experiments to compare with ReAct-only self-training
> > > > ___
> > > >
> > > > The following results depict the Table 4, with the task performances of the ReAct-only self-training method added. We have omitted the prior work baselines for simplicity. The best results are listed in **bold**, and the second best results are listed with [brackets].
> > > >
> > > > | Test Trial         | Round | Method     | Valid.    | Test.     |
> > > > |--------------------|-------|------------|-----------|-----------|
> > > > | Single             | 0     | ReAct-only | 69.1 / 43.2 | 72.4 / 45.8 |
> > > > | Single             | 1     | ReAct-only | 69.2 / 43.2 | 73.1 / 47.0 |
> > > > | Single             | 2     | ReAct-only | 70.0 / 43.5 | **74.3** / 47.8 |
> > > > | Single             | 3     | ReAct-only | **70.5** / **44.0** | [73.9] / 47.2 |
> > > > | Single             | 0     | Ours       | [70.1] / 41.0 | 72.4 / 45.4 |
> > > > | Single             | 1     | Ours       | 69.7 / 43.0 | 73.1 / **49.0** |
> > > > | Single             | 2     | Ours       | 69.0 / [43.8] | 73.0 / [48.0] |
> > > > | Single             | 3     | Ours       | 69.1 / 42.8 | [73.9] / **49.0** |
> > > > |
> > > > | Iterative (accum.) | 1     | ReAct-only | 73.9 / 49.7 | 76.1 / 50.4 |
> > > > | Iterative (accum.) | 2     | ReAct-only | 75.8 / [52.0] | 77.4 / 52.6 |
> > > > | Iterative (accum.) | 3     | ReAct-only | **76.6** / **53.5** | [78.0] / [53.4] |
> > > > | Iterative (accum.) | 1     | Ours       | 74.0 / 47.3 | 76.6 / 51.6 |
> > > > | Iterative (accum.) | 2     | Ours       | 75.1 / 49.5 | 77.8 / [53.4] |
> > > > | Iterative (accum.) | 3     | Ours       | [75.9] / 51.3 | **78.2** / **54.8** |
> > > >
> > > > According to the results, the performance of the ReAct-only method becomes better on the validation scenarios across different rounds, demonstrating the effectiveness of the contrastive self-training method used to train the models in Rounds 1~3. However, it is shown that our framework achieves the better success rate on test scenarios, and the consistency of outperformance is further demonstrated in the iterative refinement evaluation setting. The reason might be that the trajectories gathered with ActRe enjoy higher scenario-wise **Success Coverage** and **Outperformance Rate** compared with those by ReAct-only sampling, and overfitting is thus reduced when training with the data gathered with ActRe.
> > > >
> > > > We hope the added experiments can further address your concerns, and we will add these experiments as well as relevant discussions to the paper.
> > > >
> > > >
> > > >
> > > > ### Other weaknesses
> > > > ___
> > > >
> > > > ***Weakness 2. Lack of methodological contributions.***
> > > >
> > > > As mentioned in our paper, the primary contribution of our work is the closed-loop framework for agents to self-improve. In this framework, agents autonomously annotate the rationales for the sampled actions from the environment in each trajectory, and self-improve through contrastive self-training. While we indicated in Section 2.2 that we follow the idea of policy gradient for self-training, we conduct the rearrangement in Eq.(2), which has inspired us with its relationship to SFT and contrastive training (Remarks 1~3). We have also conducted corresponding ablation studies in Sec. 4.1 to validate the relationship and the superiority of contrastive self-training. As indicated in Sec. 4.1, we leave the integration of more advanced RL techniques for future work.
> > > >
> > > > ***Weakness 3. Diversity and scalability issues of agent trajectories.***
> > > >
> > > > Our ActRe allows a ReAct agent to directly sample actions from the environment by annotating the rationales in each trajectory. By grounding in the environment, our framework can in theory collect trajectories with arbitrary action allowed in the environment at each time step. Furthermore, the sampling process of ActRe can be executed in parallel for different trajectories, since the probability at each time step to sample an action and query ActRe is fixed as $p=0.5$ (Appendix A).
> > > >
> > > > According to the comparison of the collected trajectories (see the augmented Table 6), the trajectories collected by ActRe are of consistently higher reward and success rate than ReAct-only sampling that relies on the internal reasoning of LLMs only. Besides, Reflexion suggested in FireAct does not yield improved performance on WebShop because of the environmental complexity (indicated by the Reflexion paper in Appendix B.1). These experiments and facts validate that our framework collects agent trajectories with improved diversity and scalability. We will add relevant discussions to our paper.

---

### Official Review · Reviewer_aDjj · 2024-05-16

**Rating:** 7
**Confidence:** 3
**Ethics Flag:** 1

**Summary:**

This paper presents $A^3$T, a framework that enables the autonomous Annotation of Agent Trajectories in the style of ReACT. The idea is to modify the trajectories with an ActRe prompting agent, which would give the reasoning for any actions. In this way, the ReAct-style agent executes multiple trajectories for specific tasks and can select successful ones together with failed ones for contrastive self-training.
The contrastive self-training is realized by policy gradient with binarized rewards. Empirical results show that the method can achieve significant improvements compared with existing techniques and methods.

**Questions To Authors:**

- How do you choose $t$ when you do synthetic data/trajectories generation?

**Reasons To Accept:**

- This paper presents a simple yet effective method for generating synthetic trajectories without human annotations/supervision.
- The policy improvement method includes both successful and unsuccessful trajectories, which I think would be the key to improving performance (similar to DPO).
- The authors conducted experiments on AlfWorld and Webshop and achieved better empirical performance.

**Reasons To Reject:**

- My major concern is that the authors only conduct experiments on two tasks. It would be great to show the advantage over other method on more tasks or benchmarks such as toolbench or toolqueries.

---

> ### Author Rebuttal · Authors · 2024-05-31
>
> Thank you for your supportive comments!
>
> ***Weakness 1. Experiments with more tasks and benchmarks.***
>
> Thank you for the suggestions! Due to time and budgetary limits, we plan to experiment with more tasks and benchmarks in the future.
>
> ***Question 1. Choice of t when synthesizing trajectories.***
>
> In our framework, at each time step, the finetuned ReAct agent determines whether to (i) rely on its own reasoning-then-action or to (ii) sample a novel action and query ActRe for reasons with probability $p=0.5$ (Appendix A). Therefore, the $t$ that ActRe is queried with a sampled action is randomly chosen in a trajectory with $p=0.5$.

---

### Official Review · Reviewer_Db4p · 2024-05-25

**Rating:** 7
**Confidence:** 3
**Ethics Flag:** 1

**Summary:**

This work introduces the A3T framework, which autonomously annotates agent trajectories in the ReAct style using an ActRe prompting agent. The primary idea behind this work is to have an agent that explains the reasons behind arbitrary actions, allowing the ReAct-style agent to synthesize new trajectories by combining these explanations with sampled actions.

The paper has been clearly written, with the motivation and solution fleshed out in reference to the existing works. The paper clearly identifies the challenge of collecting multi-step reasoning and action trajectories, and emphasizes the issues of autonomous decision-making.

Under extensive experiments on AlfWorld and WebShop, the authors clearly show improvements over ReAct, and conduct numerous ablation studies to study the different components of their proposed algorithm. I believe, the findings in the paper will be valuable to the community.

**Reasons To Accept:**

The paper's strength lies in its clean exposition to the existing literature and the issues that need to be addressed for completely autonomous agents with LLMs. The closed-loop system for continuous self-improvement is a standout feature. A3T iteratively refines the agent’s performance through autonomous annotation and contrastive self-training. The validation of the framework through comprehensive experiments in different environments (AlfWorld and WebShop) shows its robustness and generalizability.

**Reasons To Reject:**

As-such, I don't have many concerns with the papers.

---

> ### Author Rebuttal · Authors · 2024-05-31
>
> Thank you for your supportive review! We will continue to contribute to the community by improving our work.

---

### Official Review · Reviewer_PnAD · 2024-05-26

**Rating:** 6
**Confidence:** 3
**Ethics Flag:** 1

**Summary:**

This paper proposes A3T, a framework for autonomous annotation of agent trajectories in the style of ReAct, which enables contrastive self-training of language agents. The key component of A3T is the ActRe prompting agent, which provides textual rationales for arbitrary external actions. The ReAct-style agent leverages ActRe and environmental feedback to synthesize trajectories for self-training. The contrastive self-training process utilizes policy gradient methods with binarized rewards to improve the task success rate. The authors conduct experiments on AlfWorld and WebShop, demonstrating the superiority of A3T over multiple strong baselines, including prompting with GPT-4, advanced agent frameworks, and fully fine-tuned LLMs.

**Reasons To Accept:**

- The proposed A3T framework addresses the scalability issue in collecting expert demonstrations and the limitations of implementing diverse agent frameworks for trajectory annotation. By leveraging the ActRe prompting agent, A3T enables autonomous trajectory annotation, reducing human effort and proprietary LLM calls.
- The contrastive self-training process in A3T utilizes policy gradient methods with binarized rewards, which effectively boosts the task success rate. The ablation studies demonstrate the significance of this technique compared to supervised fine-tuning and policy gradient with original rewards.

**Reasons To Reject:**

- While the authors mention the potential of incorporating advanced RL and RLAIF algorithms into A3T as future work, the current study does not explore these directions. Investigating the integration of more sophisticated RL techniques could further enhance the performance and sample efficiency of the A3T framework.
-  The initial trajectory set for Round-0 training in A3T is obtained by ReAct prompting with gpt-3.5-turbo-instruct-0914. While this provides a starting point for the self-training process, it may limit the diversity of the initial trajectories. The paper does not explore alternative methods for generating the initial trajectory set, such as using different prompting techniques or combining trajectories from multiple sources. Investigating the impact of the initial trajectory set diversity on the final performance could provide insights into the importance of this factor and potential ways to improve it.

---

> ### Author Rebuttal · Authors · 2024-05-31
>
> Thank you for your valuable suggestions!
>
> ***Weakness 1. Lack of investigation of more sophisticated RL techniques.***
>
> As stated in our paper, the major contribution of our work is the closed-loop framework for agents to self-improve, where agents directly sample actions from the environment, autonomously annotate the rationales, and self-improve by contrastive self-training. We leave the integration of more advanced RL techniques for future work.
>
> ***Weakness 2. Explorations of alternative methods for generating the initial trajectory set to improve its diversity.***
>
> We first consider alternative prompting techniques like Reflexion, which is also suggested by the FireAct [1] work. However, as indicated by the Reflexion paper [2] in Appendix B.1, Reflexion does not yield improved performance on WebShop because of the environment complexity.
>
> To improve the diversity of the initial trajectory set, we also experiment with ReAct-only sampling for multiple times, and assign the same API calls budget as that for ActRe in Round 1 of our framework. After collection, ReAct-only sampling obtains Reward=$81.6$ and Success Rate $=55.3$%. The quality of these gathered trajectories is better than that of the initial trajectory set in the paper (Table 6, Round 0: Reward=$68.5$ and Success Rate $=40.0$%). After the initial SFT process in Round 0, the trained ReAct agent obtains **R**/**SR**=$72.4$/$45.8$% on test tasks, while ours in Round=0 achieves **R**/**SR**=$72.4$/$45.4$% on test tasks.
>
> This suggests a better agent initialization could be achieved by providing more diverse initial trajectories. However, this comes at the cost of more API calls, and the cost would be higher when integrating trajectories from other sources, *e.g.*, human annotators. In contrast, the main purpose of our work is to propose the closed-loop framework for agents to annotate trajectories *autonomously*, and we have demonstrated the effectiveness of the framework when starting from a limited initial trajectory set. Due to time and budgetary limits, we will add more diverse trajectories into the initial set, and conduct more comprehensive studies on it in the future. We will also add the related discussions to our paper.
>
> [1] Chen et al., 2024. FireAct: Toward Language Agent Fine-tuning.
>
> [2] Shinn et al., 2023. Reflexion: Language Agents with Verbal Reinforcement Learning.

---

### Decision · Program_Chairs · 2024-07-10

**Decision:**

Accept

**Comment:**

Summary:
The paper introduces A3T, a framework for autonomously annotating agent trajectories in the ReAct style using an ActRe prompting agent. It enhances self-training for language agents by explaining actions and synthesizing new trajectories. A3T surpasses strong baselines in experiments on AlfWorld and WebShop, demonstrating its effectiveness in improving autonomous decision-making for agents.

Overall, the concerns raised by the authors on additional experiments comparing with ReAct, alternatives for initial trajectories, diversity and scalability of the trajectories are well-addressed. ChatGPT. Their experiments also demonstrate that ActRe achieves a broader coverage of high-reward trajectories compared to sampling with ReAct alone.

While the technique is not very novel, the idea of A3T has a lot of significance in handling scale and providing a new experimental base on closed loop solution for the future work.

Pros:
1. The experiments highlight the importance of this approach compared to supervised fine-tuning and policy gradient with non-binarized rewards.
2. The A3T framework handles scalability issues in gathering expert demonstrations and implementing diverse agent frameworks for trajectory annotation.

Cons:
1. The technique is not very novel as such, as it is very similar to REINFORCE.

Novelty:
This work presents a closed-loop system for continuous improvement, where A3T iteratively enhances agent performance through autonomous annotation and contrastive self-training. Extensive experiments validate the framework's robustness and versatility.

Significance:
1. The paper presents a thorough exploration of existing literature and the obstacles in developing completely autonomous agents with LLMs. A3T uses policy gradient techniques with binary rewards in its contrastive self-training process to significantly improve task success rates. With this motivation, the authors present a closed loop solution with ActRe which opens up a lot of promising avenues to explore RL techniques in this paradigm extensively in the future.
2.  It uses the ActRe prompting agent for autonomous trajectory annotation, reducing human input and dependence on proprietary LLM calls.

[comments from the PCs] Please add the ablations and comparison you conducted during the discussion period to the paper. Also, please conduct more detailed analyses on the types of trajectories that are sampled initially leading to SR and diversity of actions. This can help understand where the improvements are coming from.